# Evaluation of Osteochondritis Dissecans Treatment with Bioabsorbable Implants in Children and Adolescents

**DOI:** 10.3390/jcm11185395

**Published:** 2022-09-14

**Authors:** Łukasz Wiktor, Ryszard Tomaszewski

**Affiliations:** 1Department of Trauma and Orthopaedic Surgery, Upper Silesian Children’s Health Centre, 40-752 Katowice, Poland; 2Department of Trauma and Orthopedic Surgery, ZSM Hospital, 41-500 Chorzów, Poland; 3Faculty of Science and Technology, Institute of Biomedical Engineering, University of Silesia in Katowice, 40-007 Katowice, Poland

**Keywords:** osteochondritis dissecans, OCD, children, adolescent, bioabsorbable implants

## Abstract

(1) Background: This is the first systematic review concerning the treatment of osteochondritis dissecans with the use of bioabsorbable implants. The study was done as a comprehensive review to identify important factors affecting the results of OCD treatment in children and adolescents; (2) Methods: We searched electronic bibliographic databases including PubMed, Cochrane Library, Scopus, and Web of Knowledge until May 2022. This systematic review was performed according to PRISMA (Preferred Reporting Items for Systematic Reviews and Meta-Analyses) and PICO (Patients, Interventions, Comparisons, Outcomes) guidelines; (3) Results: We identified 2662 original papers of which 11 were found to be eligible for further analysis. The study group included a total of 164 OCD lesions in 158 patients. In 94.86% of postoperative cases, there was complete healing or local improvement on follow-up CT or MRI scans. The great majority of patients achieved a good clinical effect. Out of 164 OCD lesions, 10 did not heal (6.09%); (4) Conclusions: Surgical treatment of stable and unstable OCD in children with the use of bioabsorbable implants facilitates a high rate of healing and a good clinical outcome; treatment of juvenile OCD is associated with a better outcome compared to adult OCD; the use of bioabsorbable implants for the treatment of humeral capitellum OCD is associated with a more frequent incidence of synovitis (18.2%).

## 1. Introduction

Osteochondritis dissecans (OCD) is a condition describing the focal separation of an articular cartilage and subchondral bone from the remaining articular surface. It affects boys 2–4 times more than girls and is most common between 10 and 20 years of age [1,2]. The prevalence ranges from 9.5 to 29 cases per 100,000; however, it could be underestimated because some cases are asymptomatic and are found incidentally [1,3]. Typical locations include the knee, talus, and humeral capitellum. OCD mostly affects the knee with approximately 85% of lesions occurring in the medial femoral condyle [4]. The etiology remains unclear and poorly understood; genetic, traumatic, and vascular causes have all been considered [4]. OCD can be classified as “juvenile”, when it is diagnosed before physeal closure, and as “adult” in skeletally mature patients. Symptoms can be minor and nonspecific, or more disturbing, such as severe pain, swelling or locking, especially in patients with free loose bodies, [5].

Correct diagnosis of OCD requires a combination of precise physical examination and imaging studies (MRI stands as the gold standard). Some cases require additional invasive diagnostics during arthroscopic joint inspection. Treatment methods depend on the patient’s age and the size, location and stability of the lesion; methods include trial of non-weightbearing for 3–6 months, cast immobilization, bone marrow stimulation techniques such as transarticular or retroarticular drilling, lesion fixation (open or using arthroscopic techniques) and, least often, excision of fragments [5,6,7]. Multiple implants are used for OCD fixation, including K-wires, cannulated screws, Herbert screws, bone pegs and bioabsorbable pegs and screws. Bioabsorbable fixation devices have been used for years to treat small fractures, osteochondral and chondral lesions [7,8,9]. Recently, a lot of effort has been directed at using more biologic methods of fixation. Therefore, bioabsorbable devices have gained in popularity especially for treating unstable OCD lesions [10].

There is a certain inaccuracy in the prognosis for OCD. Based on the literature, it is thought that juvenile OCD has a better prognosis than adult OCD. Stable lesions in skeletally immature patients have an excellent prognosis when treated nonoperatively. Surgical treatment is indicated for unstable lesions and after nonoperative treatment failure.

On the contrary, according to many authors, juvenile and adult OCD are very similar regarding the prognosis but discovered at different points of skeletal maturity [6,7,8,9]. The main limitation of the literature studies of bioabsorbable fixation of OCD in the pediatric population is that they are limited to small series. The aim of this study was to review the literature and evaluate the OCD cases treated with bioabsorbable implants. An additional purpose was to interpret whether the surgical treatment of stable and unstable OCD in children repaired via a bioabsorbable fixation device provides healing and a good clinical outcome.

## 2. Methods

### 2.1. Literature Searches and Study Selection

This systematic review was performed according to PICO (Patients, Interventions, Comparisons, Outcomes) and PRISMA (Preferred Reporting Items for Systematic Reviews and Meta-Analyses) guidelines. The PubMed, Cochrane Library, Scopus, Web of Knowledge databases were searched. Search queries used were (“osteochondritis dissecans”) AND (“children” OR “adolescent”) AND (“bioabsorbable implants”). No filter was used. Two authors screened the results. The initial search identified 2662 results. After introductory revision, 24 papers were chosen for further analysis. 11 papers met the inclusion criteria and were included in the study. The study inclusion scheme is presented in a flowchart—Figure 1.

### 2.2. Inclusion and Exclusion Criteria

We used PICO framework-based research questions for this review. PICO criteria used in the study are shown in Table 1. If articles met predefined criteria, they were included. The criteria for eligibility for selection of a paper were (1) published in English, (2) patients aged 18 and younger and (3) stable and unstable OCD lesions treated surgically with bioabsorbable implants. The exclusion criteria for this review were (1) nonhuman studies, (2) not original studies, (3) full text not available, (4) traumatic osteochondral lesions, (4) previous surgery, (5) concomitant lesions or additional surgical procedures that may affect the outcome and (6) review articles.

### 2.3. Data Extraction and Outcome Measures

Demographic data, OCD stage (MRI and arthroscopic classification systems), indications for surgery, symptoms, type of implant, follow up, final effect and complications were carefully extracted. Risk of bias was carefully assessed and double-checked by two authors using Robvis Cochrane tool/ROB2 (Figure 2 and Figure 3) [11]. Great variation in MRI classification systems used in the reviewed articles was observed, including the Hefti, ICRS, Dipaola, Hughes, Nelson and De Smet systems. The same problem was observed for arthroscopic classification systems, including the Guhl, Ewing and Voto, and ICRS systems.

### 2.4. Statistical Analysis

In the statistical analysis, considering that there are various methods of assessing the quality of variable coherence allowing the extraction of certain relationships between variables applicable in medicine [12,13], Spearman’s rho correlation coefficients were used to compare the relationships between various parameters published by different authors. In order to verify whether men and women differ in OCD incidence, the statistics of the Wilcoxon test were calculated. Only unit-consistent results were compared.

## 3. Results

The study group included a total of 164 OCD lesions in 158 patients. There were 113 boys (71.5%) and 45 girls (28.5%). The mean age of patients was 13.69 years, and the mean follow-up was 32.43 months. The study included 89 immature patients with open physes and 39 patients with closed physes. There were 30 patients from the Adachi et al. [14] group; although the study concerned “juvenile OCD”, we couldn’t find the data on whether all of these patients were immature with open physes.

The most common OCD location was MFC—95 cases, then LFC—31 cases, humeral capitellum—22 cases, patella—10 cases, tallus—3 cases and patellar groove—2 cases. Of the 93 patients with available data, the mean size of OCD was 289.13 mm^2^. OCD dimensions of the remaining patients are summarized in Table 2.

The number of bioabsorbable implants (n) used for OCD stabilization depended on lesion size and surgeon experience. The distribution was checked with the Shapiro–Wilk test; it was statistically significant (W = 0.981; *p* = 0.972) and consistent with the normal distribution. The median was 3.00; therefore notwithstanding the mean, we could conclude that 50% of the implants were at least *n* > 3.

The average number of bioabsorbable implants used for OCD stabilization was 3 (1–11) depending on lesion size and surgeon experience. In 8 studies, the authors used bioabsorbable pins for OCD stabilization (136 lesions), in 2 studies bioabsorbable screws (27 lesions) and, in last case, one bioabsorbable pin plus one screw. Details are shown in Table 3.

The analysis included 32 OCD lesions that were stable on MRI (Hefti grade II) that qualified for surgery after conservative treatment failure (after 3 months for the Tabador et al. [15] group and 6 months for the Komnos et al. [16] group) and 132 OCD lesions that were unstable on MRI scans. There were 22 cases of detached OCD in the study group. A compilation of OCD grades according to the MRI classification with grades according to the intraoperative classification (if available) is presented in Table 2. We emphasize that only patients with reduction and ostechondral stabilization with bioabsorbable implants were analyzed; all additional procedures affecting the treatment outcome were excluded from the review. In only one case we found a mechanical implant complication, a broken biodegradable magnesium-based pin. In eight cases, there was a prolonged joint effusion after surgery due to synovitis (knee—four cases; elbow—four cases). Considering the total number of knee joints, synovitis after surgery occurred in 2.8% of cases. For the elbow joint, synovitis occurred in 18.2% of cases. No patient had the implant dislodged or showed destabilization causing secondary damage to adjacent cartilage. In total, 94.86% of postoperative cases showed complete healing or local improvement on follow-up CT or MRI scans; moreover, the great majority of patients achieved a good clinical effect. Treatment results, including healing rates, are presented in the Table 4. Out of 164 OCD lesions, 10 did not heal (6.09%). Failure cases are presented in the Table 5.

**Table 2 jcm-11-05395-t002:** The study group overview (concerns were marked—*; not available—NA).

Author	Study Period	Study Group	Male/Female	Open/Closed Physes	Age [Year]	Follow-Up [Month]	OCD Stage MRI Class	OCD Stage Arthroscopic Class
**Ronga et al.** **[17]**	NA	1	1/0	1/0	11	24	1 grade IV ICRS	NA
**Tabaddor et al.** **[15]**	2000–2006	24	14/10	6/18	14.4 (11–16)	39.6 (19–74)	11 grade II; 12 grade III; 1 grade V Hefti	13 grade II; 10 grade III; 1 grade IV Ewing and Voto
**Takeba et al.** **[18]**	NA	4	4/0	2/2	14.5 (12–16)	6 (3–7)	4 grade III ICRS	NA
**Camathias et al.** **[19]**	2005–2009	13 patients/16 OCD	12/4	13/0	12.3 (11–15)	27 (10–53)	2 grade II; 3 grade III; 11 grade IVa Hughes	4 grade I; 7 grade II; 5 grade III Guhl
**Adachi et al.** **[14]**	2002–2010	30 patients/33 OCD	23/7	30/0 NA * (juvenile?)	14.4 (11–17)	39.6 (25.2–75.6)	17 grade III; 16 grade IV Nelson’s	NA
**Galagali et al.** **[20]**	NA	1	0/1	1/0	14	10	1 grade IV ICRS	1 grade IV Ewing and Voto
**Chun et al.** **[21]**	2007–2014	11	10/1	2/9	16.3 (11–19)	51 (12–91)	4 grade II; 7 grade III Dipaola	5 grade II; 6 grade III Guhl
**Jungesblut et al.** **[22]**	2018–2021	9 out of 19 patients	3/6	4/5	14.22 (12–16)	11.44 (6–20)	9 grade III ICRS	NA
**Komnos et al.** **[16]**	2004–2016	40 * (up to 16 yo)	28/12	40/0	13.1 (11–16)	79.2 (36–156)	21 grade II; 19 grade III Hefti	21 grade II; 19 grade III Guhl
**Zeilinger et al.** **[23]**	2014–2016	7	3/4	7/0	12.1 (10–16)	29.9 (7–49)	3 grade 2; 3 grade III; 1 grade IV Dipaola	3 grade I; 2 grade II; 1 graed III; 1 grade IV Guhl
**Uchida et al.** **[24]**	2006–2009	18	18/0	13/5	14.28 (12–16)	39 (36–50)	7 grade II; 9 grade III; 2 grade IV De Smet	5 grade II; 11 grade III; 2 grade IV ICRS

**Table 3 jcm-11-05395-t003:** The study group overview (continuation). Not available—NA.

Author	MFC	LFC	Patella	Patellar Groove	Humeral Capitellum	Tallus	OCD Size Overall	Number of Implants	Pin Type
**Ronga et al.** **[17]**	-	-	-	1 entire	-	-	550 mm^2^	3	poly-lactic acid pins (SmartNail, Bionix Implants, Tampere, Finland)
**Tabaddor et al.** **[15]**	14	5	5	-	-	-	257 mm^2^ (40–900)	2.3 (1–7)	poly-96L/4D-lactide copolymer pins (SmartNail, ConMed, Linvatec, Finland)
**Takeba et al.** **[18]**	-	-	-	-	4	-	62.75 mm^2^ (40–96)	4 (3–5)	poly-L-lactide absorbable pins (GBFDÒ, Stryker, Japan)
**Camathias et al.** **[19]**	14	2	-	-	-	-	244 mm^2^ (50–961)	2 (1–3)	poly-96L/4D-lactide copolymer (SmartScrews, ConMed Linvatec, Finland)
**Adachi et al.** **[14]**	16	11	4	2	-	-	427.9 mm^2^ ± 197.2	3.4 (1–9)	bioabsorbable pins (NEOFIX, Gunze, Kyoto, Japan)
**Galagali et al.** **[20]**	-	1	-	-	-	-	160 mm^2^	1 pin; 1 screw	NA
**Chun et al.** **[21]**	7	4	-	-	-	-	319 mm^2^ (120–500)	at least 2/NA	poly-L-lactic acid screws (Arthrex, Naples, FL, USA)
**Jungesblut et al.** **[22]**	6	-	-	-	-	3	292.44 mm^2^ (60–532)	3.66 (2–6)	magnesium-based pins (MAGNEZIX Pins, Hannover, Germany)
**Komnos et al.** **[16]**	33	7	-	-	-	-	grade II 18.5 (14–26 mm) grade III 20.3 (14–28 mm)	2.3 (1–4)	poly L-Lactide pins (SmartNail, ConMed, Linvatec, NY, USA)
**Zeilinger et al.** **[23]**	5	1	1	-	-	-	3800 mm^3^ (200–20,200)	4.7 (3–9)	hydroxyapatite/poly-L-lactic acid pin (u-HA/PLLA, Osteotrans^®^, Takiron Co Ltd., Osaka, Japan)
**Uchida et al.** **[24]**	-	-	-	-	18	-	NA	3.1 (1–5)	HA/PLLA thread pins (Super Fixorb 30-thread pin; Takiron Co., Ltd., Osaka, Japan)

**Table 4 jcm-11-05395-t004:** Treatment results overview (concerns were marked—*; not available—NA).

Author	Clinical Outcome	MRI Findings	Recovery Rate	General Complication	Implants Complication	Synovitis
**Ronga et al.** **[17]**	improvement from 90 to 95 Lysholm score	complete healing on MRI	100%	none	0	0
**Tabaddor et al.** **[15]**	improvement from 7.3 to 7.9 Tegner activity score * (2 patients lower score)	interval or complete healing in 15 out of 17 MRI * 17 out of 24 had MRI	87.5%	2 out of 24 non-healed needed restabilisation (1 more non-healed with no symptoms-no reoperation)	0	2/24
**Takeba et al.** **[18]**	NA; good short-term results	3 out of 4 healing on CT; 1 improvement on CT	100%	none	0	0
**Camathias et al.** **[19]**	improvement from 1.43 to 3.44 Hughston score	NA	100%	2 out of 16 (transient peroneal nerve neurapraxia due to concomitant meniscal repair; saphenous nerve neurapraxia)	0	2/16
**Adachi et al.** **[14]**	improvement from 80 to 96 Lysholm score	32 out of 33 complete healing on MRI	96.96%	none	0	0
**Galagali et al.** **[20]**	NA; 4 Hughston score	compleat healing on CT	100%	none	0	0
**Chun et al.** **[21]**	improvement from 32.6 to 82.8 Lysholm score	11 out of 11 improvements on MRI (imrovement from 2.6 to 1.27 Dipaola score)	100%	none	0	
**Jungesblut et al.** **[22]**	NA	8 out of 9 complete healing on MRI (1 non-healed)	88.88%	1 needed second surgery	1 pin broken	0
**Komnos et al.** **[16]**	improvement from 70.4 to 95.1 Lysholm score	36 out of 40 complete healing on MRI (4 non-healed)	90%	4 needed second surgery	0	0
**Zeilinger et al.** **[23]**	improvement from 13 to 15 Ogilvie-Harris scale	6 out of 7 complete healing on MRI (imrovement from 3 to 1 Dipaola score)	85.7%	1 out of 7 non-healed needed restabilisation (OCD 20.2 cm^3^)	0	0
**Uchida et al.** **[24]**	improvement from 68 to 98.06 Mayo Elbow Performance Index	15 out of 18 healing with mean MOCART score of 87	94.44%	1 out of 18 needed second surgery (loose body)	0	4/18

**Table 5 jcm-11-05395-t005:** Failure cases overview. Not available—NA.

No	Sex	Physes Status	Age [Year]	OCD Stage/MRI	OCD Stage/Arthroscopy	OCD Location	OCD Size	Number of Implants	Revision Surgery
1.	female	partially closed	14	Hefti II	Ewing II	MFC	16 × 12 mm	1	yes/further fixation
2.	male	partially closed	15	Hefti II	Ewing III	LFC	30 × 30 mm	2	no/pain free
3.	female	closed	17	Hefti V	Ewing IV	MFC	15 × 15 mm	2	yes/further fixation
4.	female	open	12	ICRS III	NA	MFC	19 × 28 mm	4	yes/repeated OCD drilling
5.	male	open	11.6	Hefti II	NA	MFC	19 mm	2	yes/autologus bone grafting + HA scaffold
6.	male	open	12.1	Hefti III	NA	MFC	20 mm	3	yes/autologus chondrocyte implantation
7.	female	open	13	Hefti III	NA	LFC	24 mm	3	yes/autologus chondrocyte implantation
8.	female	open	13	Hefti III	NA	MFC	24 mm	3	yes/autologus chondrocyte implantation
9.	NA	closed	16	Dipaola IV	Guhl III	MFC	43 × 47 × 10 mm	4	yes/refixation with metal screws
10.	male	open	14	IV ICRS	NA	humeral capitellum	10 × 8 × 4.5 mm	3	yes/lesion removal

After applying the principal components method with Varimax factor rotation, two factors were identified. All analyzed variables explained 63.96% of the total variance. It was shown in the first construct that the recovery rate was better if there were fewer patients with closed physes in the study (Table 6). On the other hand, the second construct showed an association between fewer implants used for stabilization and fewer patients with open physes (Table 6). Based on Spearman’s rank correlation coefficient it has been proven that there is no statistical correlation between OCD size and recovery rate (Rho = −0.191; *p* = 0.651) and there is no statistical correlation between number of implants used and recovery rate (Rho = −0.479; *p* = 0.136). Spearman’s rank correlation coefficient has shown that there is statistical correlation between the ratio of open to closed physes and recovery rate: Rho = −0.470; *p* = 0.144 for patients with open physes and Rho = −0.209; *p* = 0.538 for patients with closed physes.

Figure 4 shows the dispersion between the analyzed variables.

## 4. Discussion

The crucial finding of this review is that the surgical treatment of stable and unstable OCD in children with the use of bioabsorbable implants is a good method that facilitates a high rate of healing and a good clinical outcome. For patients with nonoperative treatment failure on stable OCD lesions and patients with unstable lesions the healing rate was 94.86%. Our review is in line with a recent, large case series, of 47 adolescent patients treated with bioabsorbable implants by Bradley at al., which reports 87.2% of patients returning to full activities or permitted to return to full activities [25]. That study was excluded from our review because it contained OCD cases and osteochondral fractures with no clear demarcation between them. There is no gold standard on treating OCD lesions, furthermore, the definition of OCD stability is not precisely defined. Moreover, the natural history of untreated OCD, like the prognosis of treated OCD, is also poorly defined. However, it is confirmed that, if lesions persist despite an adequate nonoperative treatment they have the potential to detach [5]. The period of time destined for nonoperative management was variable and ranged from 3 to 6 months for the included studies. The heterogeneity of the length of time for clinical improvement before proceeding to surgery constitutes a limitation of this review. Magnetic resonance imaging is currently considered the gold standard for the evaluation of OCD. However, inconsistency of preoperative OCD classifications within a variety of MRI classification systems is another limitation. This heterogeneity, together with the lack of standardization in OCD classification and the eligibility for surgical treatment makes comparative assessment challenging and mainly limited to a descriptive assessment. Many techniques for the treatment of OCD are described. Antegrade or retrograde OCD drilling has been well documented in the literature, but as standalone procedures they are reserved for stable lesions. Kocher et al. reported 100% efficacy of retrograde transarticular drilling in immature patients [9]. Furthermore, Anderson et al. reported that antegrade drilling gave an 83% healing rate in skeletally immature patients [26]. While reviewing the literature, there were also reports that drilling alone had no influence on the OCD outcome [2]. Such dissonance about treatment results may raise doubts in the system of decision making for appropriate treatment method.

Unstable lesions require the internal fixation of the osteochondral fragment in order to improve the healing [2]. Many kinds of metal implant are used for OCD stabilization, including cannulated screws, Herbert screws and metal staples [27,28,29,30,31].

There are many reports of both skeletally mature and immature patients with very good OCD treatment results (more than 90% healing rate) using metal implants [28,29,30]. A great advantage of metal devices is the possibility of early postsurgical mobilization with implementation of the rehabilitation program. Metal implants are associated with MRI interference, which makes noninvasive assessment of the healing process difficult. In our opinion, this is the main factor which should limit the use of metal implants in the OCD treatment. Although they provide good stability, there are also some mechanical complications associated with these implants, such as loosening or damage requiring additional surgical procedures [28]. That is why the use of bioabsorbable implants in the treatment of OCD becomes so important and has recently come into favor. The main advantages include gradual load transfer to bone during the implant resorption and no need for subsequent implant removal. There are also some concerns about less rigid fixation with bioabsorbable pins which could eventually lead to nonunion [32]. Based on studies designed for adult patients, bioabsorbable pins and screws provide both OCD healing and symptom relief [7,33,34,35]. 

With reference to inaccuracy in the prognosis for OCD treatment, based on the data analysis in our study, we conclude that treatment of juvenile OCD is associated with better outcome (Rho = −0.209; *p* = 0.538) and requires fewer implants for fixation. Unfortunately, there are no studies directly comparing different kinds of bioabsorbable implants. Weckstrom et al. compared outcomes in young adults treated with bioabsorbable nails (73% healing) with pins (35% healing) [36]. This confirms how important compression is, and not just the stabilization of the OCD lesion. All implants used in the reviewed studies allowed for compression, which probably corresponds with high healing rate. On the other hand, use of bioabsorbable implants is not without drawbacks. The most frequently reported one is synovitis, which is related to the host response to the polymer’s biodegradation [29,37,38]. The synovitis is more often associated with polyglycolic acid (PGA) implants as a result of the rapid degradation of the biomaterial, in contrast to polylactic acid (PLA) implants that degrade slowly [39]. We noticed eight cases of synovitis, four in the knee and four in the elbow.

Confirmed knee synovitis has been associated with SmartNail (ConMed, Tampere, Finland) which is a copolymer comprised of both PGA and PLA. The goal of this combination is to reduce the complications associated with the inflammatory response related to PGA’s rapid degradation and the complications associated with the slow degradation of PLA.

Uchida et al. reported joint effusion in 4 out of 18 patients (24%) which was significantly higher for the elbow compared to other locations [24]. They used SuperFixorb 30-thread pins (Takiron Co., Ltd.) which are the composite of hydroxyapatite (HA) and poly-L-lactide acid (PLLA).

This combination is believed to ensure much greater initial mechanical strength than PLLA alone and has osteobinding and osteoconductive potential [40]. In our opinion, high effusion rate in the elbow may constitute a limitation of the use of bioabsorbable material in the treatment of humeral capitellum lesions. Use of a low number of bioabsorbable implants, ensuring the OCD stabilization, or use of metal implants should be considered. In addition, implant destabilization with back-out, implant breakage and OCD nonunion has been described [37,41,42,43,44,45,46,47]. Some authors emphasize the possibility of uneven resorption of the implant which can lead to back-out of articular implant part, resulting in intra-articular loose bodies and chondral abrasions [33,48,49,50]. In our review, we observed only one mechanical complication in the form of a pin breakage. Camathias et al. reported, altogether, 14 broken screws (SmartScrew; ConMed Linvatec) of a total of 61 implants in 24 patients (breakage rate −23%) based on MRI follow-up. Interestingly, not all patients were symptomatic and only four patients underwent surgical revision for implant failure during the follow-up [43]. In our review, we included the earlier work of this author from 2011, because the paper describing complications published in 2015 lacked data relevant for analysis, which distorts the frequency of bioabsorbable implants complications.

The treatment of OCD lesions should be relatively straightforward and should be taken at the right time, preferably before detachment of a loose body, in order to avoid specialist techniques like autologous chondrocyte implantation (ACI) or osteochondral grafts. Further analysis is needed to understand the impact of lesion size on outcomes after bioabsorbable fixation. A head-to-head study comparing metal to bioabsorbable fixation implants may bring some important findings.

## 5. Conclusions

Surgical treatment of stable and unstable OCD in children with the use of bioabsorbable implants facilitates a high rate of healing and a good clinical outcome.The use of bioabsorbable implants allows for a non-invasive control of the OCD healing process.Treatment of juvenile OCD is associated with a better outcome and requires fewer implants for fixation compared to adult OCD.The use of bioabsorbable implants for the treatment of humeral capitellum OCD is associated with a statistically more frequent incidence of synovitis (18.2%).

## Figures and Tables

**Figure 1 jcm-11-05395-f001:**
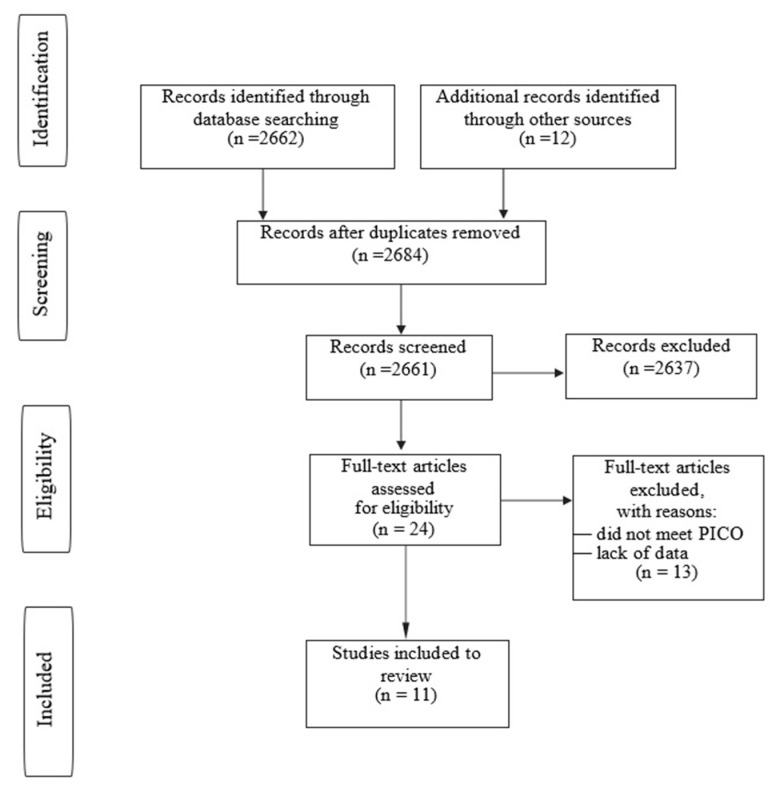
Flowchart showing PRISMA protocol for data acquisition.

**Figure 2 jcm-11-05395-f002:**
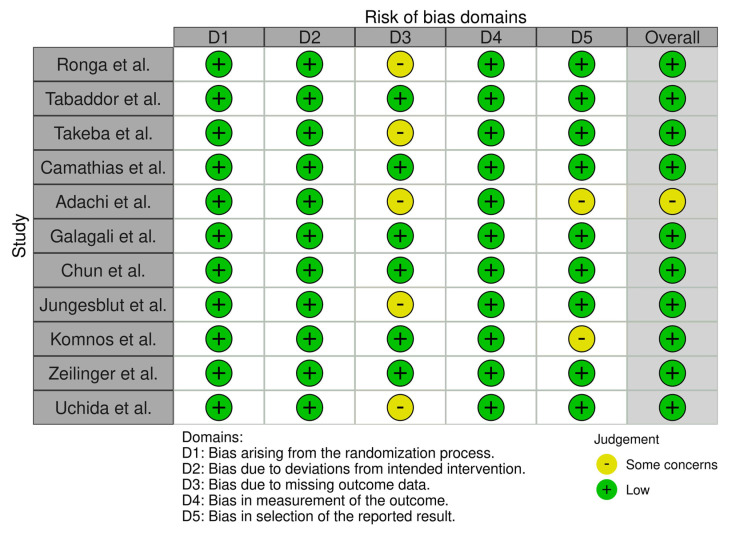
“Traffic light” plots of the domain−level judgements for each individual result.

**Figure 3 jcm-11-05395-f003:**
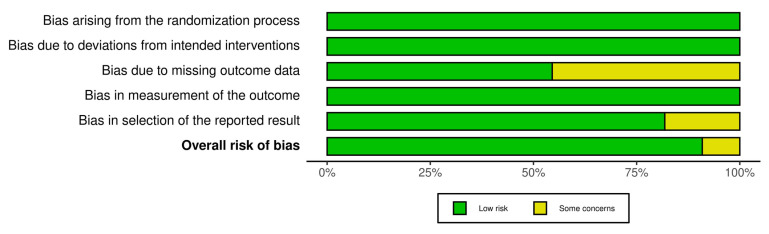
Weighted bar plots of the distribution of risk−of−bias judgements within each bias domain.

**Figure 4 jcm-11-05395-f004:**
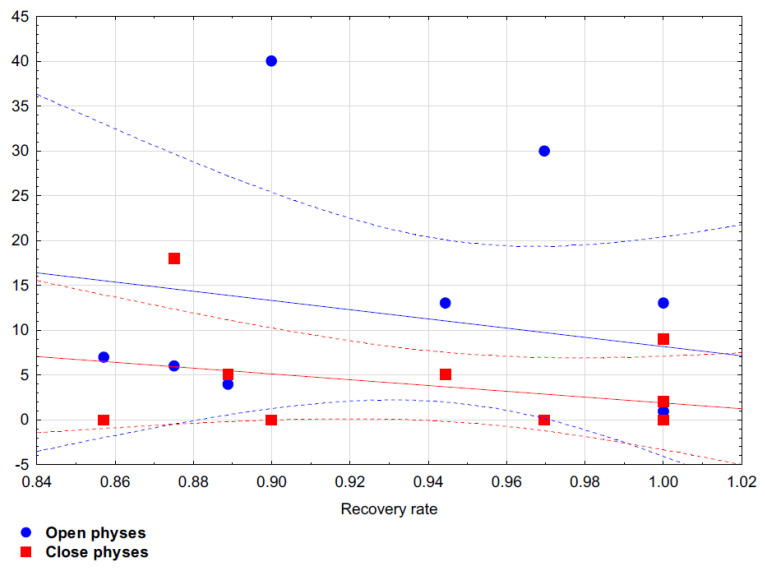
Dispersion of results with a dashed confidence line for results between recovery rate and open and closed physes.

**Table 1 jcm-11-05395-t001:** PICO criteria used in the study.

PICO	Description
**Patients**	Patients 18 years old and younger with osteochondritis dissecans (each location).
**Intervention**	Surgical treatment with bioabsorbable implants.
**Comparisons**	The diagnostic, clinical and local results of treatment in patients with open and closed physes.
**Outcomes**	The outcomes and usefulness of bioabsorbable implants depending on the location and stage of OCD.

**Table 6 jcm-11-05395-t006:** Result of principal components analysis.

	Factor 1	Factor 2
Number of implants	−0.106	−0.711
Recovery rate	0.912	0.219
OCD size overall	−0.060	−0.630
Open physes	0.188	−0.696
Closed physes	−0.917	0.199

## Data Availability

The data generated and analyzed in the current study are available from the corresponding author on reasonable request.

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
