# Peer review of "Evaluation of Osteochondritis Dissecans Treatment with Bioabsorbable Implants in Children and Adolescents"

_jcm, 2022, doi:10.3390/jcm11185395_

Round 1

Reviewer 1 Report

Dear Authors

Thank you for the possibility to read and revise your manuscript entitled “Evaluation of osteochondritis dissecans treatment with bioabsorbable implants in children and adolescent”.

The manuscript is interesting and provides a comprehensive overview of the outcomes of treating osteochondritis dissecans, shedding light on the clinical value of using bioabsorbable implants in children and adolescents. The risk of bias has been thoroughly assessed, the Results are presented in a clear way and the Discussion is appropriate. However, I have some minor issues that I think you should address to improve your work. Please, in general, take care also of some minor mistakes in English writing.

·       In the Abstract, the sentence “to identify important clinical findings in treatment children and adolescents” is grammatically incorrect.

·       In the “Literature Searches and Study Selection” paragraph you inappropriately switch from past to present tense, which makes the reading less clear.

·       Please correct Figure 1. “Records screened” and “Records after duplicates removed” should be inverted.

·       In Figure 4 correct “psyses” with “physes”.

·       In the Results, this sentence doesn’t make sense: “30 patients from the Adachi et al [13] group, although the study concerned "juvenile OCD", was not specified whether all patients had an open physes.” Furthermore, please use the word physis when it is singular; “physes” is a plural noun.

·       You state that the average number of bioabsorbable implants used for OCD stabilization was 3 (1-11), but this type of data has often a non-normal distribution. Therefore, mean would not be the appropriate measure of central tendency to represent your findings. Please check it and, in case, modify.

Author Response

Dear reviewer.

We would like to kindly thank you for your time spent reviewing our manuscript ‘’ Evaluation of osteochondritis dissecans treatment with bioabsorbable implants in children and adolescent. Systematic review’’. We appreciate all your valuable comments of our work. We have revised our manuscript, according to your suggestions. We believe that the manuscript has been further improved. All revisions made were marked up using the “Track Changes” function in MS Word so changes can be easily viewed.

1) In the Abstract, the sentence “to identify important clinical findings in treatment children and adolescents” is grammatically incorrect.

We corrected the sentence as recommended: ‘’The study was done as a comprehensive review to identify important factors affecting the results of OCD treatment in children and adolescents’’.

2) In the “Literature Searches and Study Selection” paragraph you inappropriately switch from past to present tense, which makes the reading less clear.

We corrected grammar errors as recommended: ‘’ The initial search has identified 2,662 results. After introductory revision, 24 papers were chosen for further analysis. 11 papers met the inclusion criteria and were included in the study. Study inclusion scheme is presented in a flowchart – figure 1’’.

3)Please correct Figure 1. “Records screened” and “Records after duplicates removed” should be inverted.

We improved the flowchart as recommended.

4) In Figure 4 correct “psyses” with “physes”.

We corrected the error.

5) In the Results, this sentence doesn’t make sense: “30 patients from the Adachi et al [13] group, although the study concerned "juvenile OCD", was not specified whether all patients had an open physes.” Furthermore, please use the word physis when it is singular; “physes” is a plural noun.

We corrected the sentence as recommended: ‘’ 30 patients from the Adachi et al [13] group, although the study concerned "juvenile OCD", we couldn't find the data whether all patients were immature with an open physes’’.

6) You state that the average number of bioabsorbable implants used for OCD stabilization was 3 (1-11), but this type of data has often a non-normal distribution. Therefore, mean would not be the appropriate measure of central tendency to represent your findings. Please check it and, in case, modify.

We modified as recommended: ‘’ The number of bioabsorbable implants (n) used for OCD stabilization depended on the lesion size and surgeon experience. The distribution was checked with the Shapiro Wilk test. Statistically significant (W = 0.981; p = 0.972) it is consistent with the normal distribution. The median was 3.00, apart from the mean, we could conclude that 50% of the implants were at least n> 3’’.

All authors have approved the manuscript changes and agree with its submission to Journal of Clinical Medicine.

Best regards.

Łukasz Wiktor

Reviewer 2 Report

Title: Evaluation of osteochondritis dissecans treatment with bioabsorbable implants in children and adolescent.

Journal: J. Clin. Med.

The objective of this research was conducted to investigate the literature and evaluate the OCD cases treated with the use of bioabsorbable implants.

The approach is original. The manuscript reads smoothly and is easy to understand. The aims, scope, and results of the study are clearly stated. I have very much enjoyed reading this paper. I find it interesting and clearly written. The study provides a very valuable addition to this line of research, and adds relevantly to the subject with additional original findings. I thus find that this paper definitively delivers results that will surely be of interest to the readership of the “J. Clin. Med.” The authors must develop the introduction that seem very poor and to develop and resume the limitations of the literature’s studies.

Author Response

Dear reviewer.

We would like to kindly thank you for your time spent reviewing our manuscript ‘’ Evaluation of osteochondritis dissecans treatment with bioabsorbable implants in children and adolescent. Systematic review’’. We appreciate all your valuable comments of our work. We have revised our manuscript, according to your suggestions. We believe that the manuscript has been further improved. All revisions made were marked up using the “Track Changes” function in MS Word so changes can be easily viewed.

The authors must develop the introduction that seem very poor.

We developed the introduction section including limitations of the literature as recommended (in the response we present only a new text): ‘’ Correct diagnosis of OCD requires a combination of precise physical examination and imaging studies (MRI stands the gold standard). Some cases require additional invasive diagnostics during arthroscopic joint inspection. Treatment methods depend on the patient age, lesion size, its location and stability including trial of non-weightbearing for 3–6 month, cast immobilization, bone marrow stimulation techniques such as transarticular or retroarticular drilling, lesion fixation (open or using arthroscopic techniques) and least often excision of fragments [5-7]. Multiple implants are used for OCD fixation, like K-wires, cannulated screws, Herbert screws, bone pegs and bioabsorbable pegs and screws. Bioabsorbable fixation devices have been used for years to treat small fractures, osteochondral and chondral lesions [7-9]. Recently, a lot of effort is directed at using more biologic methods of fixation. Therefore, bioabsorbable devices have gained in popularity especially for treating unstable OCD lesions [10]. There is certain inaccuracy in the prognosis for OCD. Based on literature it is thought that juvenile OCD have a better prognosis than adult one. Stable lesions in skeletally immature patients have an excellent prognosis when treated nonoperatively. Surgical treatment is indicated for unstable lesions and after nonoperative treatment failure. In contrary, according to many authors juvenile and adult OCD are very similar regarding the prognosis but discovered at different points of skeletal maturity [6-9]. The main limitation of the literature studies of bioabsorbable fixation of the OCD in the pediatric population is that they are limited to small series. The aim of the study was to review the literature and evaluate the OCD cases treated with bioabsorbable implants. Additional purpose was to interpret whether the surgical treatment of a stable and unstable OCD in children repaired via a bioabsorbable fixation device provide healing and a good clinical outcome’’.

All authors have approved the manuscript changes and agree with its submission to Journal of Clinical Medicine.

Best regards.

Łukasz Wiktor